# Hierarchical Clustering of DNA k-mer Counts in RNAseq Fastq Files Identifies Sample Heterogeneities

**DOI:** 10.3390/ijms19113687

**Published:** 2018-11-21

**Authors:** Wolfgang Kaisers , Holger Schwender, Heiner Schaal 

**Affiliations:** 1Department of Anaesthesiology, HELIOS University Hospital Wuppertal, University of Witten/Herdecke, Heusnerstr. 40, 42283 Wuppertal, Germany; 2Institut fur Virologie, University Hospital Düsseldorf, Heinrich Heine University Düsseldorf, 40225 Düsseldorf, Germany; schaal@uni-duesseldorf.de; 3Mathematisches Institut, Heinrich-Heine-Universität Düsseldorf, 40225 Düsseldorf, Germany; schwender@math.uni-duesseldorf.de

**Keywords:** DNA-kmer, Fastq, RNAseq

## Abstract

We apply hierarchical clustering (HC) of DNA k-mer counts on multiple Fastq files. The tree structures produced by HC may reflect experimental groups and thereby indicate experimental effects, but clustering of preparation groups indicates the presence of batch effects. Hence, HC of DNA k-mer counts may serve as a diagnostic device. In order to provide a simple applicable tool we implemented sequential analysis of Fastq reads with low memory usage in an R package (seqTools) available on Bioconductor. The approach is validated by analysis of Fastq file batches containing RNAseq data. Analysis of three Fastq batches downloaded from ArrayExpress indicated experimental effects. Analysis of RNAseq data from two cell types (dermal fibroblasts and Jurkat cells) sequenced in our facility indicate presence of batch effects. The observed batch effects were also present in reads mapped to the human genome and also in reads filtered for high quality (Phred > 30). We propose, that hierarchical clustering of DNA k-mer counts provides an unspecific diagnostic tool for RNAseq experiments. Further exploration is required once samples are identified as outliers in HC derived trees.

## 1. Introduction

DNA k-mer’s are DNA subsequences of length *k*
(k∈N). The analysis of DNA k-mer counts constitutes the (word) frequency based methods of *alignment free sequence comparison* which are applied in wide areas of DNA sequence analysis (see [1,2,3] and references therein).

### 1.1. Analysis of k-mer Counts

Word based analysis of DNA sequencing data is utilised for isoform quantification in RNAseq (Sailfish [4]), genome assembly (Velvet [5] or Celera [6,7]), detection and correction of sequencing errors [8,9] and metagenomics (CLARK [10]). The quality of sequence assembly can be improved by k-mer based filtering of reads [11,12].

Counting of small word sizes (k≤10) can be done offhand, whilst analysis of longer motifs usually requires elaborate strategies due to computational and storage demands. Standard approaches in this area include hash tables (Jellyfish [2]), bloom filter based methods (BFCounter [13]) and disk based implementations (KMC [14], KMC2 [15] and DSK [16]), but numerous other methods have been described [17].

### 1.2. Quality Control of RNAseq Data

Analysis of RNAseq data usually includes quality control of raw reads, which includes counting of DNA k-mer’s [18]. FastQC [19], a widely used quality control software, for example indicates over-represented k-mer’s in a Fastq file by counting 7-mers on 2% of the contained reads in order to indicate expansive read duplication.

The comparison of DNA k-mer spectra over multiple samples for quality control of RNAseq data was initially described in a GEUVADIS report [20]. Subsequently, a *k-mer Profile Analysis Library* (kPAL), implemented in Python, has been published [21].

### 1.3. Word Sizes

The counting of DNA k-mer’s results in vectors of length $N^{n}\left(n=4^{k}\right)$ indicates, that $4^{k}$ words exist. Small values of *k* (2≤k≤4) diminish discrimination capabilities of counting results. At intermediate values of *k* (9≤k≤15), unique k-mer’s and *nullomers* (absent k-mer’s) begin to appear as a consequence of the limited complexity of the human genome [21]. At larger values of *k* (k>30), the k-mer count vectors become sparse (as only a small fraction of k-mer’s are present) [22].

Word sizes (k) used in DNA k-mer analysis depend on application type, for example k≈ 15–30 in metagenomics [10], k≈ 20–80 in genome assembly [23], k≈20 isoform quantification [4].

In comparison of DNA k-mer spectra over multiple RNAseq samples, the word size was k=9 in the initial analysis [20] and k=12 using smoothed k-mer count profiles [21].

The word size determines the specificity of the sampled sequences. While k-mers from small *k* presumably will be present in almost all sequencing samples, long k-mers (*k* > 20) become increasingly specific for species or proteins. As HcKmer analysis of all biological samples in this study was done using k=9 which can be specific for at most 3 amino acids, diagnostic criteria derived from HcKmer analysis are regarded as unspecific for presence of biological entities.

### 1.4. Analysis of DNA k-mer Counts

The vector representation $v\in N^{n}\left(n=4^{k}\right)$ of DNA k-mer counts allows application of analytic procedures provided by euclidean geometry (for example distance measures [1]) and machine learning algorithms, for example principal component analysis (PCA) or hierarchical clustering (HC).

In hierarchical clustering (HC), different entities are located in bi-parting trees according to their pairwise similarity (quantified by a distance measure). Although the trees provide no absolute measure, accumulation of biological or technical related samples in different sub-trees may reveal relevant heterogeneities. Observed sample (dis-)similarities in DNA k-mer counts have been shown to be indicative of problematic samples (for example due to read duplication or presence of rRNA) [21].

### 1.5. HcKmer Analysis Algorithm

We implemented a k-mer counting algorithm in C and provide a programming interface allowing to run the complete analysis inside R. The software is available as R package *seqTools* for download from Bioconductor. The Canberra distance is utilised as distance on DNA k-mer counts.

### 1.6. Analysed Samples

Three sample batches were downloaded from ArrayExpress (accession E-MTAB-4842, E-MTAB-4104 and E-MTAB-691), each containing two treatment groups. Second, a batch of 61 samples, sequenced in our facility and containing RNAseq data from two tissues, was analysed. The first moiety consists of 57 dermal fibroblast samples (ArrayExpress accession E-MTAB-4652); the second comprises 4 Jurkat cell samples.

## 2. Results

### 2.1. Data Collection

Fastq files from the three ArrayExpress batches were downloaded and data from each experiment was collected into one separate data-set.

The 61 samples had been sequenced in our local facility on 8 Illumina Flowcells. Data from each Flowcell was collected into one data-set; the processing of $9.8\times 10^{9}$ contained reads took 8.96 h (3.04 × $10^{5}$ reads/second) in a single thread with approximately 1 Gigabyte working memory consumption (see Section 1.1 for more details).

### 2.2. Detection of Experimental Effects

The separation of experimental groups by HcKmer is exemplified by analysis of three sequencing datasets, downloaded from ArrayExpress [24]. The first dataset has been created by RNAseq of mouse liver endothelial cells from normal (4 samples) and tumor infiltrated tissue (4 samples) [25]. Tumor growth was induced by injection of B16-F10 melanoma cells into the portal vein. Figure 1 shows a dendrogram where all samples derived from normal tissue cluster together in one sub-tree. Differential gene expression analysis revealed up-regulation of glycolytic pathways in tumor derived endothelial cells.

In the second example, RNAseq data of KBM-7 chronic myelogenous leukaemia cells was analysed. Three samples KBM-7 cells and three samples of KBM-7 cells with knockout of NUDT2 (a NUDIX hydrolase [26]) have been analysed for differential gene expression. Figure 2 shows a dendrogram where samples with and without knockout of NUDT2 cluster in different sub-trees. The analysis revealed 6288 differential expressed genes and 1685 genes with a fold-change ≥2 [27].

The third example shows analysis of Fastq files derived from whole genome sequencing. Two pairs of cell lines derived from high-grade ovarian carcinoma of two patients were analysed. Two cell lines, PEO1 obtained during the first relapse (fr) and platinum sensible (sen), and PEO04 obtained during the second relapse (sr) and platinum resistant (res) were derived from patient 2 (p2). Two cell lines, PEO14 obtained during initial presentation (pr) and platinum sensible (sen), and PEO23 obtained during relapse (re) and platinum resistant (res) were derived from patient 3 (p3). Figure 3 shows a dendrogram where samples from both patients are located in different sub-trees and additionally, platinum sensible and resistant samples also are located in different sub-trees. The analysis revealed structural variants appearing during tumor evolution, for example different tandem duplications in samples PEO14 and PEO23 [28].

### 2.3. Identification of Batch Effects

Figure 4 shows a dendrogram where 16 samples from two Flowcells (*d24a* and *c0yr*) and two cell types (dermal fibroblasts and Jurkat cells) are analysed using HcKmer. The partition tree consists of two equally sized sub-trees. Each of them only contains samples sequenced on one Flowcell although on Flowcell *c0yr*, two different cell types are present. Thus, DNA motif dissimilarity is greater between Flowcells *d24a* and *c0yr* than between fibroblasts and Jurkat cells.

Figure 5, indicates that median Phred scores in sequences from both Flowcells are sufficiently high (see also Section 1.4).

#### 2.3.1. Filtering for Sufficient Phred Scores

In order to explore, whether the observed dissimilarity can be removed, reads containing at least one Phred score < 30 were discarded. The filter was applied to Fastq files sequenced on *d24a* and *c0yr* Flowcells. From Flowcell *d24a*, 14.3% of reads in Fastq files and from Flowcell *c0yr*, 15.1% of reads in Fastq files were excluded thereby. Finally, the filtered reads were re-analysed.

In the HC-tree, the position of one Flowcell (fib13 lane03 c0yr) changed to the opposite sub-tree but still, three fibroblast samples cluster together with the Jurkat samples (Figure 6).

Thus, filtering based on Phred scores induced only a minor change in cluster formation.

#### 2.3.2. HcKmer on Sequences Aligned to the Human Genome

Fastq reads causing HcKmer tree separation may not match to the reference genome and thus would be filtered out by alignment. In order to evaluate this eventuality, clustering of raw Fastq files and mapped reads are compared. Pairs of Flowcells containing 8 fibroblast samples (*d24a*, *d10r*, *c0g9*, *c0yt*, *c2uk*, *d1pd*) are analysed by HcKmer on raw reads and on mapped reads. An example is shown in Figure 7 where raw and mapped reads cluster in similar patterns. In both trees, the size of the largest sub-tree containing samples from only one Flowcell is 7. The mean sizes of these sub-trees in all 15 possible pairs is 7.21 in raw reads and 6.57 in mapped reads. Mapping thus reduces Flowcell cluster sizes by 8.9% but still, clusters of size ≥6 are prevalent. Detailed results are shown in Section 2.

#### 2.3.3. Prevalence of Batch Effects in RNAseq Data

The prevalence of detectable batch effects was analysed on the whole set of 61 samples sequenced on 8 Flowcells. Pairwise comparison of 8 Flowcells results in 28 pairs. All Flowcell pairs are analysed for presence of batch effects using HcKmer and a semi-quantitative score ranging from strong batch effect (b1a = top-level separation of Flowcells; shown in Figure 4) and detectable batch effect (b1a, b1b, b2a or b2b) to absence of batch effects (es). From the analysed Flowcell pairs, 6 (21.4%) show strong batch effects and 17 (60.7%) show detectable batch effects. Thus, batch effects are present in a considerable fraction of Flowcell pairs. The definitions and details on analysis are shown in Section 4.

#### 2.3.4. Influence of Batch Effects on False Discovery Rate

Clustering according to sample preparation batches potentially influences differentially Expressed Gene (DEG) analysis in RNAseq data. To address this question, results from DEG analysis are compared between Flowcells with different HcKmer dissimilarity using a two-way ANOVA. Classification as batch effect (b1 or b2 vs. es) is significantly associated with an increased number of differentially expressed (DE) genes (see Section 5 for details). The ANOVA predicts 3.848 DE genes for b1a dissimilarity and 695 DE genes for idf/es (no batch effect detectable). Thus, an increased number of false positives in DEG analysis are found when batch effects are identified by HcKmer.

### 2.4. Separation Sensitivity on Simulated Data

The degree of sequence dissimilarity required to produce HcKmer trees with top level separation of two sample groups (e.g., Flowcells) is determined in simulation where Fastq files with random DNA are analysed. A pure random group is compared to a group in which a variable percentage of random DNA reads is contaminated with (one or multiple) fixed DNA 6-mers.

Clustering of groups is quantified using a score (Contralaterality Score, CS). A decrease of CS from 41.1% (mean value for pure random sequences) to 0% is regarded as indicative for clustering of groups and in the setting of the simulation, a CS < 10% is considered to be statistical significant (*p* < 0.05).

Figure 8 shows separation capabilities of HcKmer for contamination of 0–6% of Fastq reads with one fixed DNA 6-mer. The results indicate that with a contamination of 4% of Fastq reads, significant sample separation reaches a power of 80%. Details of simulation, definition of CS and results are shown in Section 3.

### 2.5. K-mer Spectrum Responsible for Tree Separation

Standard quality control tools (for example FastQC) only report a small number of over-represented k-mers which may not represent a sufficient sample for explanation of batch effects diagnosed by HcKmer. We therefore estimate how many k-mers are needed in order to evoke the observed batch effects using an example.

From the simulation data on separation sensitivity, it is assumed that ≈3% contamination (where the median CS falls below 12.5% in Figure 8) is required in order to produce a strong batch effect (b1a). Two pairs of Flowcells are selected, one with complete tree separation due to batch effect (Flowcells *d24a* and *c0yr*) and one pair of Flowcells without tree separation due to batch effect (Flowcells *d24a* and *d1pd*). For each k-mer, the logarithmised and normalised sum of k-mer counts on whole Flowcells are calculated and compared for both pairs and are shown in Figure 9). In the example shown in the left panel (*d24a*/*c0yr* comparison), differences from the 3678 k-mers with the largest difference in k-mer count need to be accumulated in order to attain a 3% contamination rate.

Thus, the observed tree separations are caused by the sum of widespread small differences instead of large disparities on a small number of k-mers. Also, in addition, the read content of problematic samples seemingly cannot be easily corrected (for example by removing or clipping a small fraction of reads).

## 3. Discussion

The underlying principle of HcKmer is the comparison of multiple samples using a high dimensional criterion which is closely linked to the central information content of DNA sequencing data. As such, this principle differs from standard quality control algorithms which analyse single samples, but also imposes restrictions, as a minimum of 3–4 samples per group are required for assessment.

Clustering of samples according to biological or experimental entities is an unspecific criterion as different k-mer spectra may be caused by differential gene expression or differing splicing patterns (as biolgically unspecific indicator, see Section 1.3) as well as the mentioned disturbing factors (read duplication or presence of rRNA [21]). Thus, etiologic factors may not be apparent.

### 3.1. Indication of Batch Effects

Transcriptome sequencing data is prone to disturbing effects resulting from experimental design for example ozone levels [29], random hexamer priming [30], GC-content [31], and transcript length [32]. Batch effects, variation due to experimental design, is a prevalent phenomenon and clustering according to surrogate values (processing date or batch) is a common approach for recognition of underlying sources of variation [33].

Analysis of our samples shows, that clustering of preparation batches may be prevalent (21–61%), and that resulting disturbances may not be removed by Phred based filters or by alignment to a genome.

HcKmer clustering according to sample preparation groups had also been demonstrated for whole genome sequencing (WGS) data as well as for whole exome sequencing (WES) data (see Figures 3 and 4 in [21]). Although unproblematic according do FastQC measures, the WGS samples exhibited significant inconsistencies in paired end read mapping suggesting, that the indicated batch effects might affect DEG analysis, which is in line with the results shown in Section 2.3.4. Thus, differences identified by HcKmer should initiate exploration of underlying effects due to potentially influential effects on downstream analysis.

Due to analysis results of differential gene expression on our fibroblast samples [34], we assume that there are no consistent differences between different groups and thus, GC-content and transcript length are unlikely to evoke the observed batch effects. Also, in addition, observed sample dissimilarities seem to result from a larger number of small differences in k-mer counts which can not be diagnosed by identification of few over- or underrepresented k-mers. Based on the shown results, our group decided to discard data from samples sequenced on two other Flowcells (data not shown here) exhibiting strong batch effects when compared with all Flowcells in the shown analysis.

### 3.2. Application of HcKmer

Standard quality control procedures address Phred-scores, GC content, over-represented K-mers and sequence duplication [18] and are implemented in ubiquitously used software products like FastQC. The shown results indicate that HcKmer represents a quality differing from qualities examined by standard procedures which essentially can be explained by the fact that between-sample heterogeneity is examined at high resolution.

A limitation is that HcKmer provides no quantitative measure for similarity and that subjective judgement is required. However as HcKmer can easily be applied (using *seqTools*), it appears advisable to include HcKmer into analysis standards.

## 4. Materials and Methods

### 4.1. Algorithmic Framework

For a DNA sequence of length *k*, $4^{k}$ different sequence motifs (k-mers) exist, each defining a category for which occurrences can be counted. The value of *k* is usually chosen in the range of 5 to 9 (resulting in 1024 to 262,144 different k-mers) as shorter motifs impair separation capabilities and longer motives increase computational demands.

The algorithms for performing HcKmer are available in R package *seqTools* on Bioconductor [35]. DNA k-mer counts are collected with other quality measures (for example Phred scores and GC content) from Fastq files. Data from multiple samples (for example from whole Illumina Flowcells) can be collected at once into merge-able objects. Merged objects then serve as input for hierarchical clustering. The Canberra distance, defined as (1)$$\begin{array}{ccc} f_{cb}\left(x_{i},y_{i}\right) & = & \frac{|x_{i}-y_{i}|}{|x_{i}\left|+\right|y_{i}|} \end{array}$$
(2)$$\begin{array}{ccc} d(x,y) & = & \sum_{i=1}^{k}f_{cb}\left(x_{i},y_{i}\right),\;{\left(x_{i}\right)}_{i=1,\ldots ,k},{\left(y_{i}\right)}_{i=1,\ldots ,k}\in R^{k}, \end{array}$$
is used as distance measure between samples. The Canberra distance (initially described in 1966 [36]), is a weighted version of the $L_{1}$ (Manhattan) metric which is sensitive for differences when both components ($x_{i},y_{i}$) are small. Additionally, total read numbers ($\sum_{i=1}^{m}x_{i}$) are scaled before calculation of Canberra distance to a common value in order to compensate a systematic offset caused by different sequencing depths (thereby increasing $|x_{i}-y_{i}|$ values). Subsequent cluster-analysis was calculated using the *hclust* function (CRAN *stats* package) [37]. Detailed source code used for the analysis is shown in Section 1.2.

#### Sample Preparation, Sequencing and Alignment

In this study, transcriptome data from 61 samples is analysed. Therefrom, 57 samples of human short term cultured human dermal fibroblasts were obtained from 27 human healthy individuals and sequenced for a study on human ageing [34]. Collection and processing of dermal samples from donors was approved by the Ethical Committee of the Medical Faculty of the University of Düsseldorf (# 3361) in 2011. The Fastq files from these samples are available under ArrayExpress accession E-MTAB-4652 (ENA study ERP015294). Also, in addition, 4 samples from cultured human Jurkat cell lines were sequenced for a study on transcriptome alterations caused by HIV infection.

Sample preparation and sequencing has been described elsewhere [34]. In short, cellular mRNA was amplificated on 8 Illumina Flowcells (v1.5) and sequenced on a Illumina HiSeq 2000 sequencer (Illumina Inc., 5200 Illumina Way, San Diego, CA, USA). From each lane, the resulting 101-nucleotide sequence reads were converted to Fastq by CASAVA 1.8.2 (Illumina sequencing analysis software). A Fastq file contained in average $162.2\times 10^{6}$ reads. In total, the 61 Fastq files contained $9.8\times 10^{9}$ reads.

Subsequent alignments were calculated on unprocessed Fastq files with *TopHat* (v 2.0.14) [38,39] using human GRCh38 assembly.

In order to compare batch effects between raw Fastq files and mapped reads, BAM file content was transformed back into Fastq (using *bam2fastq* from CRAN package rbamtools [40]). All DNA k-mer counts were collected using k=9 ($4^{9}=$ 262,144 DNA motifs), except for the simulation study (k=6).

Differential gene expression analysis was performed using Quasi-likelihood F-Tests from the edgeR (3.12.0) framework [41]. Genes with a reported FDR <0.1 were considered to be significantly differentially expressed. The same analytic procedure performed on the whole dataset resulted in no significantly differential expressed gene [34].

## 5. Conclusions

HcKmer applies an unsupervised learning algorithm onto the raw high throughput sequencing data with the capability of detecting potentially prevalent and influential unwanted variation. Experimental designs allowing HcKmer analysis (for example defined library preparation an a defined set of Flowcells) thus are favourable. When multiple samples are sequenced on the same Flowcell, the reads should be demultiplexed before HcKmer analysis. Based on contrasts identified by HcKmer, further exploration of results from sequencing experiments as well as exclusion of contaminating samples may be reasonable.

## Figures and Tables

**Figure 1 ijms-19-03687-f001:**
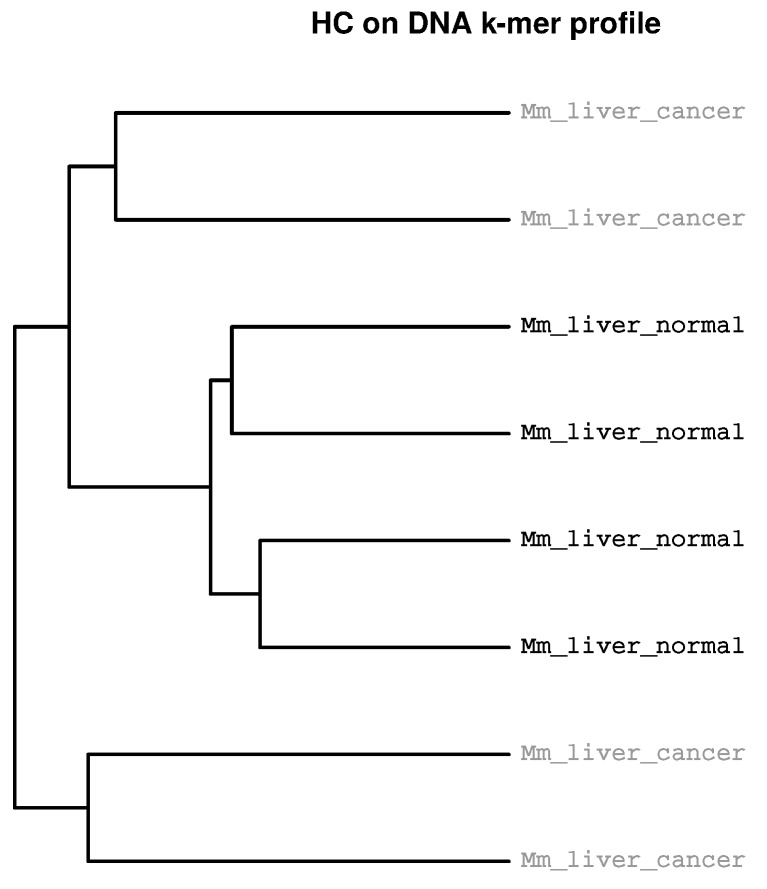
**Clustering according to experimental groups.** B16-F10 melanoma cells were injected into portal veins in mice resulting in intrahepatic tumor growth. After 14 days, endothelial cells from hepatic tumors and from livers of healthy mice were extracted and analysed with RNAseq (single end). Differential gene expression had been focused on 1255 metabolic genes expressed in endothelial cells. The sample data was downloaded from ArrayExpress (accession E-MTAB-4842).

**Figure 2 ijms-19-03687-f002:**
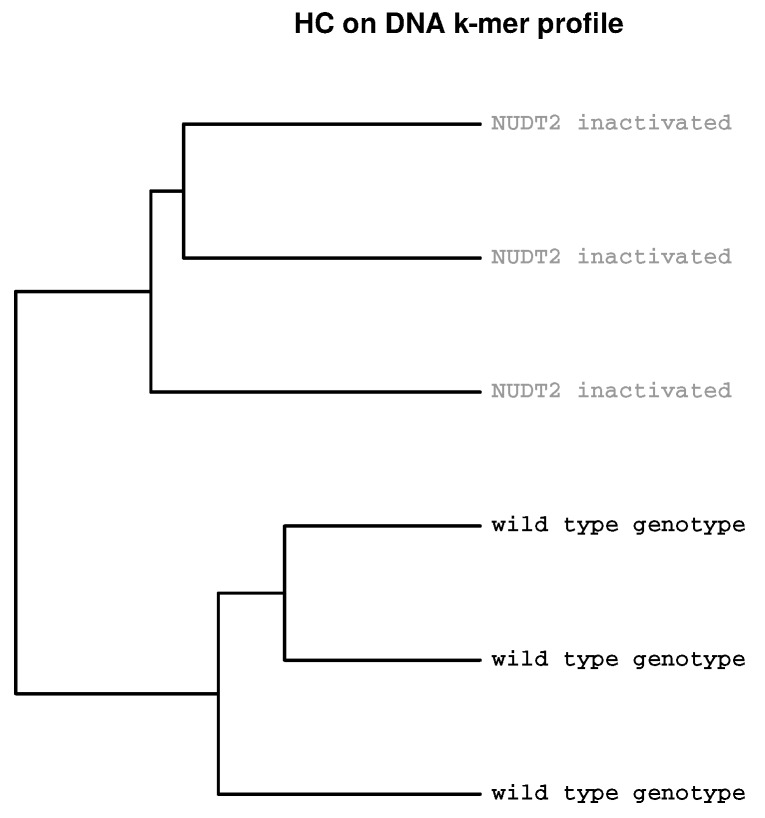
**Clustering according to experimental groups.** KBM-7 chronic myelogenous leukaemia cells with and without knockout of gene NUDT2 were analysed using RNAseq. The sample data was downloaded from ArrayExpress (accession E-MTAB-4104, only the first of the Fastq files from paired end sequencing (*_1.fastq.gz) are included).

**Figure 3 ijms-19-03687-f003:**
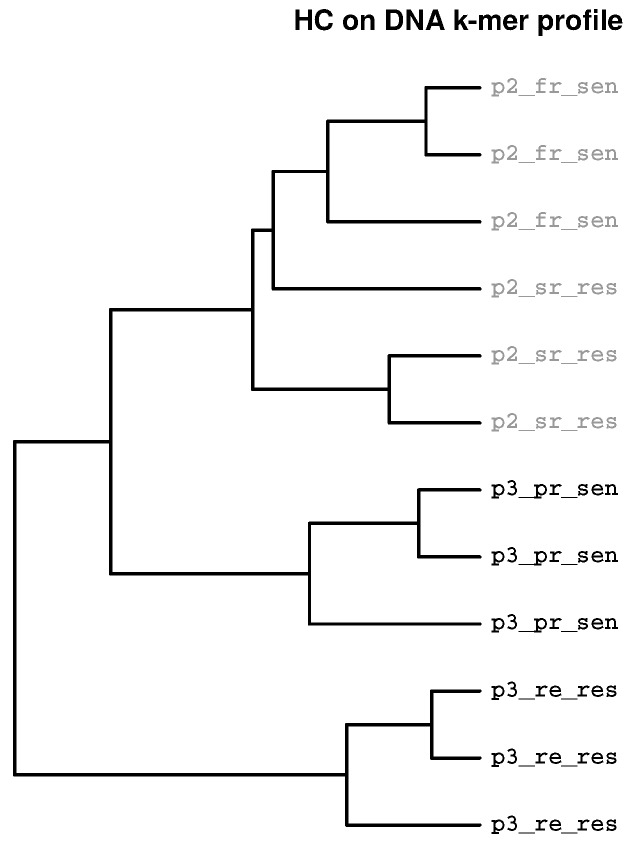
**Clustering according to experimental groups.** High grade serous ovarian carcinoma cells from two patients were characterised using whole genome sequencing. Leaf labels indicate patient (p2 and p3), clinical status (fr = first relapse, sr = second relapse, pr = primary presentation, re = relapse) and platinum sensibility status (sen = sensible, res = resistant). Samples with identical leaf labels indicate samples obtained from the same cell line. The sample data was downloaded from ArrayExpress (accession E-MTAB-691, only the first of the Fastq files from paired end sequencing (*_1.fastq.gz) are included).

**Figure 4 ijms-19-03687-f004:**
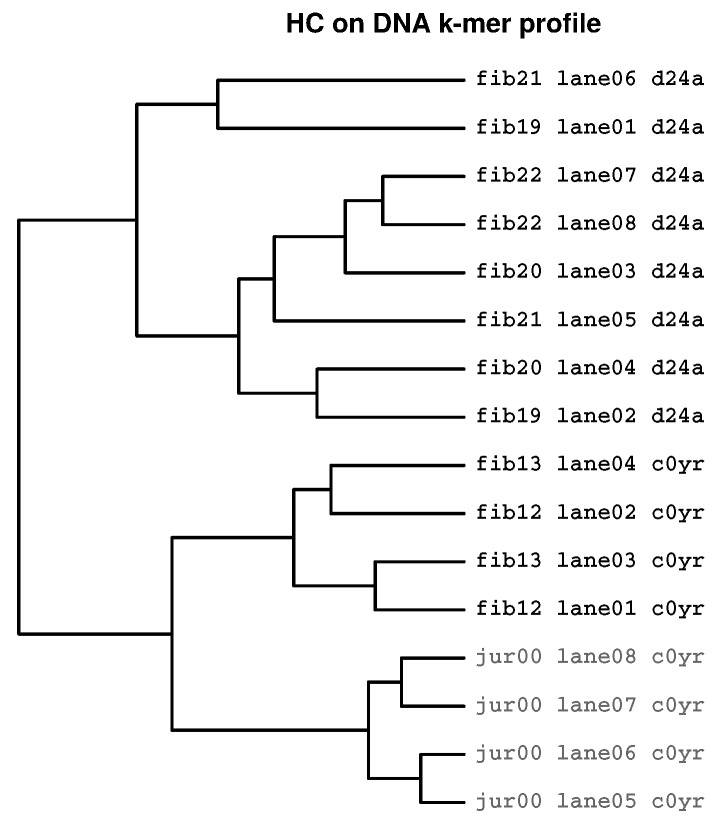
**Batch effect indicated by HcKmer on samples sequenced in two Illumina Flowcells:** Leaf labels denote cell type (fib = Fibroblasts, jur = Jurkat cells) and number of individual, lane number and Flowcell label (*d24a* and *c0yr*). Samples from Jurkat cells are highlighted in grey. The tree clearly separates Flowcell *d24a* and *c0yr* although Flowcell *c0yr* contains two different cell types.

**Figure 5 ijms-19-03687-f005:**
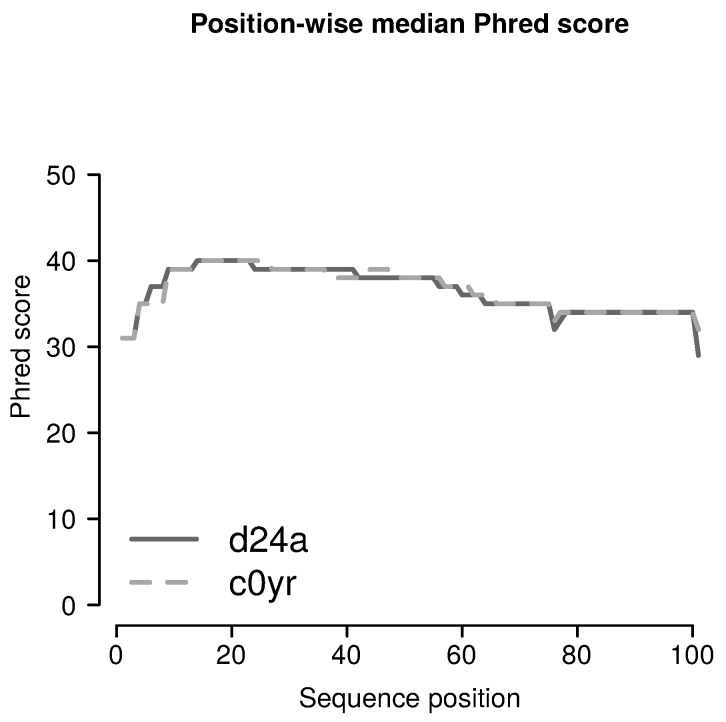
**Median Phred score values:** For each read position, median Phred scores are shown for samples sequenced on Flowcells *d24a* and *c0yr*. All median Phred scores are >28.

**Figure 6 ijms-19-03687-f006:**
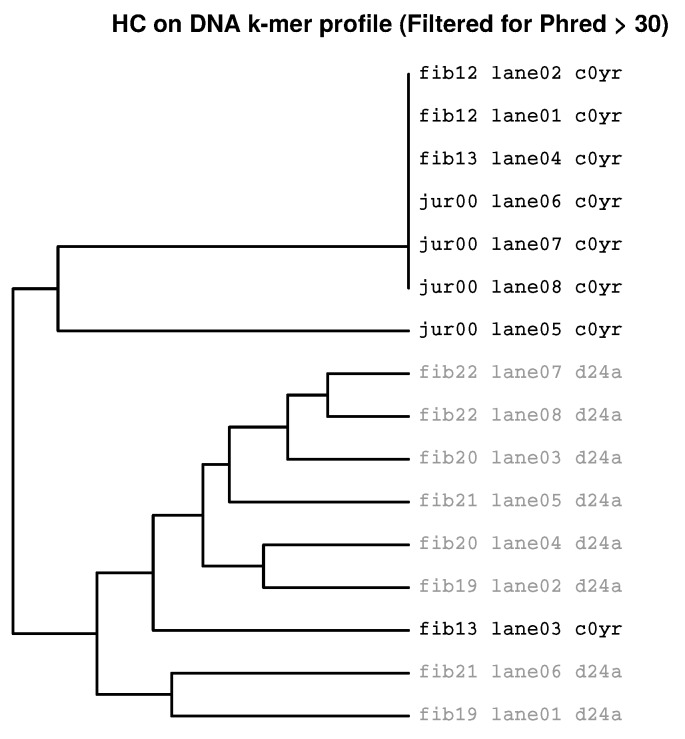
**Clustering of Fastq files containing filtered reads:** All reads containing Phred scores < 30 had been discarded before HcKmer analysis. On top level clade, the Jurkat cell samples still exclusively cluster together with samples from the same Flowcell.

**Figure 7 ijms-19-03687-f007:**
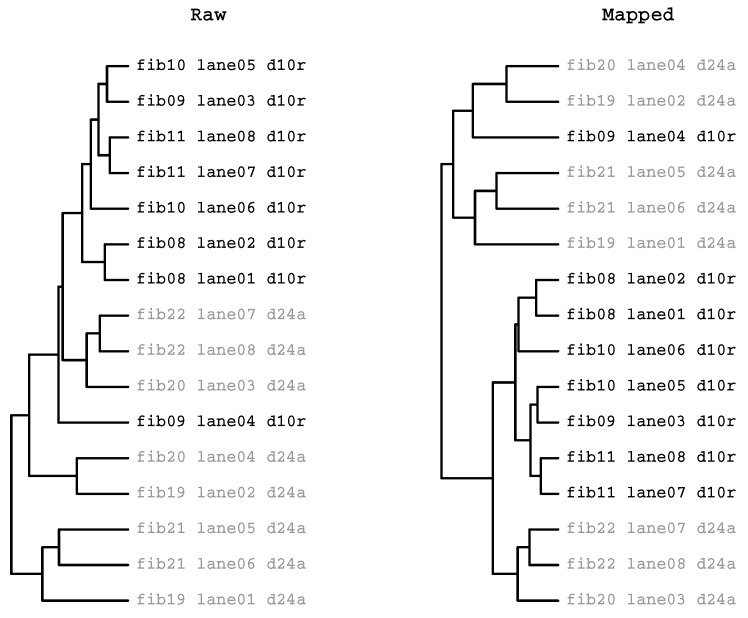
**HcKmer on raw and mapped reads:** Fastq files containing mapped reads were constructed from raw reads by alignment using TopHat followed by extraction of reads from BAM files using *reader2fastq* (rbamtools) function. HcKmer on raw and mapped reads was calculated on Fastq files from the same Flowcell pairs. Raw reads consist of unmapped and mapped reads. **Left:** HcKmer on raw reads. **Right:** Mapped reads. Leaf labels denote cell type (fib), lane number, and Flowcell label (*d24a* and *d10r*). All but one samples from Flowcell *d10r* (dark grey) cluster within a separate sub-tree. Raw and mapped reads show similar clustering characteristics.

**Figure 8 ijms-19-03687-f008:**
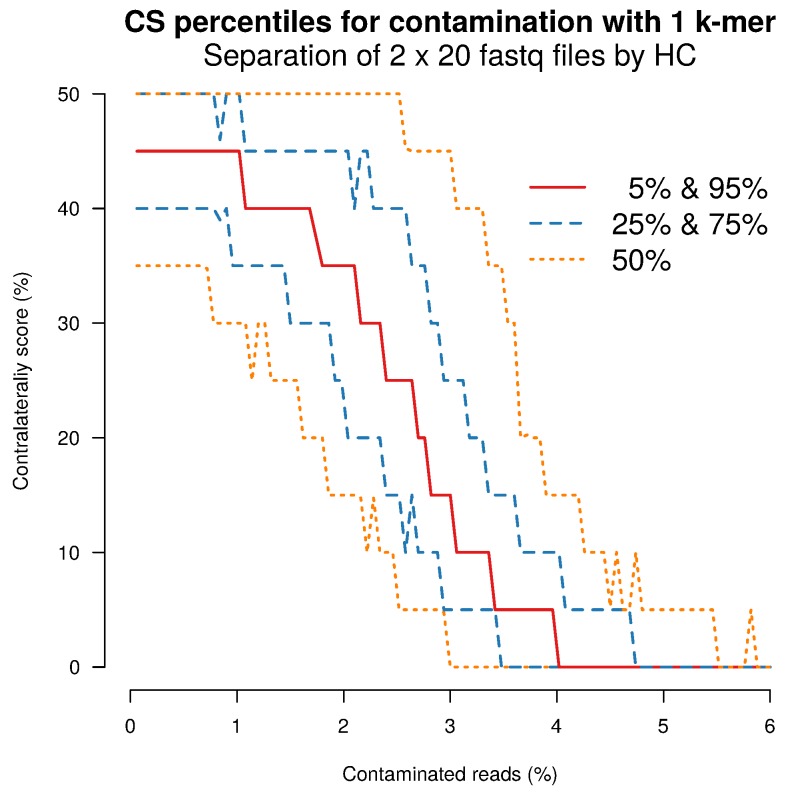
**Separation sensitivity on simulated data:** HcKmer was performed on Fastq files containing simulated DNA sequence (k=6). Percentiles for Contralaterality score (CS) are given for variable percentages of contamination with a single fixed DNA 6-mer. CS quantifies the presence of the minor present group in the first half of the HC-derived group labels. CS values <10% are considered to be statistically significant (*p* < 0.05).

**Figure 9 ijms-19-03687-f009:**
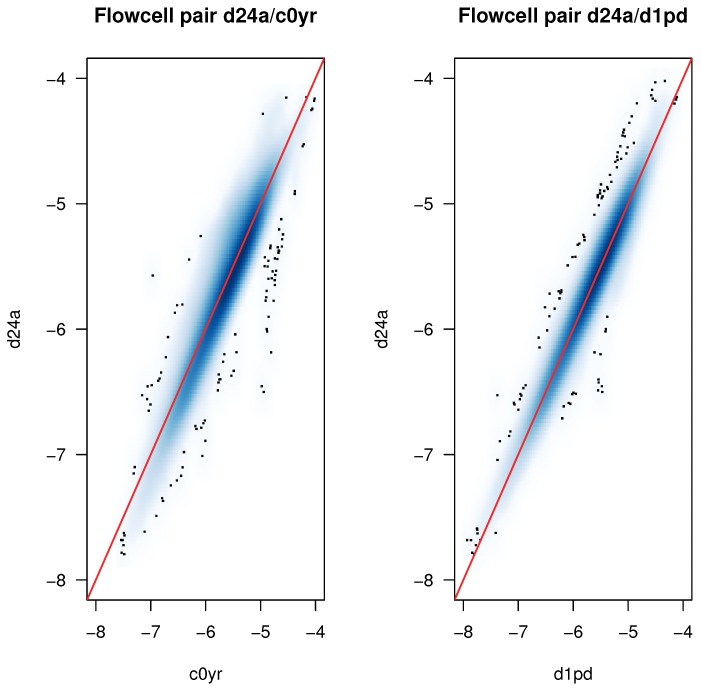
**K-mer spectrum responsible for tree separation:** Comparison of K-mer counts in two Flowcell pairs using density scatter plots: All axes represent normalised and $\log_{10}$ transformed k-mer counts on a whole Flowcell (each 8 Fastq files). The diagonal (red line) indicates equal normalised counted numbers for k-mers in both Flowcells. The luminosity of the blue area indicates point-density in the scatter plot (dark = high density). **Left panel**: Comparison of k-mer counts on Flowcells *d24a*/*c0yr*(strong batch effect (=b1a) identified by HcKmer). **Right panel**: Comparison of k-mer counts on Flowcells *d24a*/*d1pd* (no batch effect (=es) identified by HcKmer). **Result:** The k-mer count differences are larger for *d24a*/*c0yr* (mean = 0.30, sd = 0.12) than for *d24a*/*d1pd* (mean = 0.01, sd = 0.07). The HcKmer diagnosed difference of sample similarity between the Flowcell pairs is due to larger deviation of k-mer counts from the diagonal for a broad variety of k-mers (thus not generated by a small group k-mers with large deviation).

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
