# Peer review of "Hierarchical Clustering of DNA k-mer Counts in RNAseq Fastq Files Identifies Sample Heterogeneities"

_ijms, 2018, doi:10.3390/ijms19113687_

Reviewer 1 Report

Kaisers et al. describe the HcKmer tool provided in the seqTools package available from Bioconductor and that uses the distribution of kmer frequencies in raw fastq files to identify samples that are outliers among replicates for a RNAseq experiment. The manuscript is clear and well-written. The seqTools package has all the documentation expected of a Bioconductor package. SeqTools appears to have a good sized user community as evidenced by the download statistics. I was able to install the seqTools package and run example analysis without any problems.

Major revisions

1. Although the overall paper is well-written, I found the literature survey in the introduction to be somewhat lacking. Use of kmers to discriminate NGS data is not a novel concept and there are a large number of kmer-based analysis tools that have been in use for a while now. Some of these are explicitly for quality control like Rseqc (hexamers instead of 9-mers used here) and Kontaminant. Kmer analysis has wide application, for e.g. KAT that use kmers to identify the heterozygosity in a sample.

2. The authors should compare the performance and features of the seqTools package with other quality control tools available today besides FastQC.

3. The idea of hierarchical clustering of multiple error metrics into a merged object is a positive. Did the authors consider profiling the forward and reverse reads from a paired-end run separately?

4. As the number of cycles increases during synthesis, more errors start to creep in towards the 3’ end and can impact the kmer profile. A merged object for a sample with kmer and phred quality score might address this issue to some extent. Another idea would be to only sample kmers from high quality regions of a sequence to avoid the bias of low quality bases.

5. The authors state that, ”Due to this large number it is unlikely, that strong batch effects (b1a) diagnosed by HcKmer can be identified by inspecting few k-mers and that they can be eliminated by removing reads containing a small set of selected k-mers.”. Did they consider that this might be an artifact only in the limited number of samples explored in this paper?

6. I could not find a good explanation for the recommendations made in the conclusion (sequencing samples together on a defined set of Flowcells and avoid multiplexing). Can the authors please clarify? Or maybe I have misunderstood?

Minor revisions

7. A number of important analysis and figures are ensconced in supplementary data. The sections in the supplementary document should be referenced clearly in the body of the main paper to allow the reader to quickly find the relevant information.

8. I could not find the term HcKmer in the seqTools documentation. Is this intentional? It seems like the functionality is offered by other functions.

9. It will be helpful for the tools and the manuscript to use the same terminology.

10. It would be useful for users to be able to pull out the subsets of reads that do conform to other similar samples according to HcKmer profiles. Can the authors consider adding this functionality to seqTools?

11. Please update seqTools documentation to include this manuscript once it is published.

Typographical errors

12. Pg 2, Line 59: Comma after evolution

13. Pg 8, Line 119: Missing reference for FastQC

14. Pg 10, Line 184: The use of the word analysis is redundant here

Author Response

1. We added introductory remarks on the role of DNA k-mer analysis in bioinformatics technology. As the area is large, a somehow comprehensive overview would fill a whole manuscript on its own. We therefore just outline the area and provide references. During our literature research, we found the report from Anvar et.al (2014) [pmid: 25514851] which we had not seen earlier and which provides important insight into the area. We explored RSeqQC and KAT: Rseqc is Python library for quality control of RNA-seq data which includes a read_hexamer module. Some of the base functionalities (naturally) seem to be similar to the functions implemented in seqTools and some of the splice-junction functionality adresses similar questions as implementations in the R-based splice-junction analysis pipeline rbamtools/refGenome/spliceSites, we described in 2017 [pmid: 28545234]. KAT (K-mer Analysis Toolkit [pmid: 27797770]) is a multi purpose C++ library for analysis of k-mers and bases on Jellyfish2. As such, KAT is built on elaborated tools and has features (for example genome assembly analysis) which (at the moment) is out of reach of a more simplistic tool like seqTools. There are also many (other) k-mer analysis tools available for example kmerPyramid [pmid: 28633391], Turtle [pmid: 24618471], KAnalyze [pmid: 24642064], CLARK [pmid: 25879410] or khmer [pmid: 25062443].

2. We understand this suggestion. Although many reports on algorithms and implementations perform comparisons with other tools in their area, we would like to avoid this because there is much space for subjective judgement. K-mer counting with seqTools seems to be quite fast for small k (k < 12),but runtime usually is not limiting when a calculation only needs do be done once and can be performed without supervision. seqTools offers a toolchain completely in R, but this only helps users willing to work with R. We would like to keep the focus of the manuscript on the HcKmer approach, the eventually important information which may be provided and on the existence of a toolchain which can easily be installed and executed. Hopefully, this approach might also be included in other software tools.

3. We haven't analysed paired-end reads in this way up to date, but the results from Anvar [pmid: 25514851] (Fig. 3) suggest, that clustering of k-mer counts could also lead to separation of forward and reverse sequences in separate subtrees. The reversed reads may be analysed using the (seqTools) revCountDnaKmers -Function in this scenario.

4. Until now, we have not explicitly addressed the question whether k-mer mixture correlates with phred quality, but the preliminary available information (see 2.3.1 in manuscript or [pmid: 24037425, page 1017]) seems not to support this hypothesis. In fact, when HcKmer indicated disturbing effects could be removed by phred-based filtering, then the HcKmer approach would be more or less useless. In the seqTools package, there are two function implemented (fastqKmerLocs and fastqKmerSubsetLocs) which allow exploring this question.

5. The cited expression seems to be unclear. A tree separation of a sample (consisting of one or more probes) is generated by a sum of large number of differences. There may either be a small number of very large differeces or a large number of small differences causing this observation. The conclusion from the shown example is, that the latter seems to be the case. This predicts difficulties when diagnosis or correction based on a small subset of reads is desired. So, the statement is not intended to be related to the number of samples. We substituted the expression and hope, the new version is better understandable.

6. Seemingly a correct remark. We changed the sentence to "When multiple samples are sequenced on the same flowcell, the reads should be demultiplexed before HcKmer analysis". The background is, that HcKmer will not work on multiplexed data. We hope to resolve the issue thereby.

7. We added references to the manuscript.

8. Building a HcKmer tree is done in a four-step procedure: First collecting the k-mer count matrix, then merging of samples, calculation of the distance matrix and hierarchical clusterung. An example is shown in section 1.2.1 of supplemental material. Another example is present in section 7 of the package vignette. We intenionelly kept the constituents of the procedure separate from each other since this makes it easier to introduce modifications. The layout of the figures for example relies extensively on modifications of the "dendrogram" objects calculated by the "hclust" function (see the call of dendrapply inside of the plotColDend funtion (1.2.2 of supplemental material).

9. A valueable remark. We will do this later on...

10. We think, this functionality is largely covered by the "trimFastq" function in seqTools.

11. We agree that this should be done.

12. OK

13. There seems to be no reference in a scientific journal for FastQC. There is an URL in the reference list

14. We could not locate the word.

Reviewer 2 Report

a nice approach for the given problem,

1. it would also be good in the introduction to show in some phrases why this is a problem in research and how/if it is solved (=what are the
advantages of this approach)

2. "unspecific diagnostic device" as argument for this approach may be strange...if unspecific as searching tool before further research please say how and why

3. maybe you might also show in the introduction that similar techniques based on k-mer for other but still somehow not so distant problems have already been addressed as https://www.mdpi.com/2073-4425/8/4/122 or some of the papers in the literature list of this paper (just showing that k-mer composition for identification and comparison of sequences has already been used)

4. line 12 "and for" ?

what has been done is shown under 4

5. so such an somehow precise mini overview in the introduction line 25 to line 29 may be rewritten to fit better to an introduction

6. line 112

any reasons why p<0.05 is considered being significant?

https://www.ncbi.nlm.nih.gov/pmc/articles/PMC4111019/

(sorry, but just relying on this habit is nothing of value)

7. line 111

really 41,1% precisely?

(yes you get all this via supplement 3.1 eq. 2)

8. line 174 literature ?

why have you then chosen k=9 for all and why k=6 for simulatio?

even though you have written it in 4, at least where you have figures with k-mer spectrum, telling the value of k might be nice

9. under Conclusions

why is this really "an unprejudiced view"?

in general the clustering of the samples seems to fit and so your approach seems to be stable and usable

the supplement is really well written and good for a better understanding of the main paper

Author Response

1. Separation of preparation batches in HcKmer derived trees may for example indicative of read duplication or presence of rRNA in read sequences. We added a remark in section 1.4. (See also reply to next issue).

2.The term "unscecific" refers to the fact that there may be varying sources for different k-mer mixtures in sequencing data. The k-mers represent fragments of either elements of biological structure (see last paragraph of 1.3) or artifacts produced by the sequencing process. Thus underlying sources may be either experimental or artificial. Thus, when unexpected separation of samples is observed an exploration of underlying reasons should be initiated. Due to the missing link to causative elements, there is no general research problem addressed (besides a general search for batch effects).

3. As noted by Reviewer 1, k-mer bases analysis of DNA sequencing data is a large area of research and has widespread applications. We therefore have added a very short introduction into the scope of k-mer based analysis of DNA sequencing data refer to various analysis tools. We did not mention paper of Sievers et.al [pmid: 28422050] yet as they use an unnamed algorithm on viral genomes using very small k (1 to 4), so there is no direct relation to the shown work apparent.

4. We reformulated the whole sentence...

5.The requirement of a mini overwiew had also been expressed by Reviewer 1. We added introductory paragraphs into the introduction.

6. We agree, that a limit for significance of 0.05 is completely arbitrary and there is nothing specific in setting it at this point (beside the fact, that it is somehow standard in biomedical literature). We changed "CS < 10 % is considered to be statistical significant" which hopefully more precise.

7. Yes, the value can be obtained with v <- 0:20 / 20 sum(dbinom(0:20 , 20 , 0.5) * pmin(v, (1 - v))) in R.

8.We have used k=9 in order to increase sensitivity and because k=9 was used in a report from GEUVADIS [pmid: 24037425 = ref 20]. See also the new section 1.3 on word sizes. We used hexamers for simulation, because k-mer counting has been done 240,000 times for the simulation study in order to save time.
9.With "unprejudiced", we mean that the total content of the sequencing data is explored without the possibility exclude features by focussing on selected features (for example gene expression). In order to give a more specific expression, we changed the sentence to "HcKmer applies an unsupervised learning algorithm"

Round  2

Reviewer 1 Report

I am satisfied by the response and edits performed by the authors.

Reviewer 2 Report

quality of introduction and overall presentation is improved

MS is sound and all arguements are conlusive